# Simultaneous Determination of 108 Pesticide Residues in Three Traditional Chinese Medicines Using a Modified QuEChERS Mixed Sample Preparation Method and HPLC-MS/MS

**DOI:** 10.3390/molecules27217636

**Published:** 2022-11-07

**Authors:** Xuyan Fan, Tao Tang, Song Du, Ningning Sang, Hao Huang, Chenghui Zhang, Xueping Zhao

**Affiliations:** 1College of Food Science and Engineering, Hainan University, Haikou 570228, China; 2State Key Laboratory for Managing Biotic and Chemical Threats to the Quality and Safety of Agro-Products, Key Laboratory of Detection for Pesticide Residues and Control of Zhejiang, Institute of Agro-Product Safety and Nutrition, Zhejiang Academy of Agricultural Sciences, Hangzhou 310021, China; 3Key Laboratory of Tropical Fruits and Vegetables Quality and Safety for State Market Regulation, Haikou 570206, China

**Keywords:** pesticide residue, QuEChERS, MWCNTs-NH_2_, traditional Chinese medicines

## Abstract

A rapid, efficient, simple, and high-throughput method for the simultaneous determination of 108 pesticide residues in three traditional Chinese medicines (TCMs) was established, comprising an improved QuEChERS method in combination with HPLC-MS/MS based on mixed samples. A quantity of 10 mL of acetonitrile was used as extraction solvent, and 10 mg of amino-modified multi-walled carbon nanotubes (MWCNTs-NH_2_) and 150 mg of anhydrous magnesium sulfate (MgSO_4_) were selected as sorbents for dispersive solid phase extraction. The performance of the method was verified according to the analytical quality control standards of SANTE/11813/2017 guidelines. With good linearity (R^2^ > 0.9984) in the range of 2–200 μg/L for all pesticides in the selected matrices, and good accuracy, precision, and high sensitivity, the recoveries were in the range of 70–120% for more than 95% of the pesticides, with a relative standard deviation (RSD) of less than 16.82% for all. The limit of detection (LOD) and limit of quantification (LOQ) of the method were 0.01–3.87 μg/kg and 0.07–12.90 μg/kg, respectively, for *Fritillaria thunbergii* Miq (*F. thunbergii*), *Chrysanthemum Morifolium* Ramat (*C. morifolium*), and *Dendrobium officinale* Kimura et Migo (*D. officinale*). The method was successfully applied to 60 batches of actual samples from different regions.

## 1. Introduction

Traditional Chinese medicines (TCMs) have a long history, and because of their specific natural conditions and ecological environment, they are of better quality and efficacy than those produced in other regions [1,2]. Zhejiang is in a subtropical zone with a mild climate and abundant rainfall. There are many genuine medicinal materials in Zhejiang Province, among which the “Zhebawei” are the most famous; *Fritillaria thunbergii* Miq (*F. thunbergii*), *Chrysanthemum Morifolium* Ramat (*C. morifolium*), and *Dendrobium officinale* Kimura et Migo (*D. officinale*) all belong to “Zhebawei” [3]. *F. thunbergii* has the effect of clearing heat, and is an expectorant and antitussive [4,5]; *C. morifolium* has anti-inflammatory and antioxidant analgesic functions [6]; and *D. officinale* has antihyperlipidemia, antihypertensive, and antihyperuricemia properties [7]. The edible parts of these three herbs are bulbs, flowers, and stems. Due to the unique geographical environment of Zhejiang Province and the needs of the growth environment of TCMs, diseases and pests (such as root rot, gray mold, powdery mildew, aphid, and red spider) in the planting process of TCMs are prone and diverse. Chemical pesticides such as insecticides and fungicides are often used for disease and pest control, and plant growth regulators (PGRs) are also used to improve product quality and nutrition. The unregulated use of these chemical pesticides will not only result in excessive pesticide residues in TCMs, but will also have a serious impact on the efficacy of Chinese medicine and result in problems regarding factors such as food safety.

Compared with ordinary agricultural products, TCMs are complex and diverse in composition, containing a wide range of active ingredients. *F. thunbergii* contains steroidal alkaloids, fatty acids, terpenoids, saponins, amino acids, and polysaccharides [8]. The main pharmacodynamic components of *C. morifolium* include flavonoids, volatile oil, and phenylpropanoid compounds [9]. The active components of *D. officinale* include polysaccharides, flavonoids, and alkaloids [10]. These complex active ingredients play a key role in the treatment of diseases. The use of chemical pesticides will cause changes in the active components of TCMs, thus influencing their pharmacological and pharmacodynamic effects [11]. Long-term consumption of TCMs with excessive pesticide residues can also pose health risks and damage health. These pesticides may interfere with the endocrine system; lead to endocrine and metabolic disorders; produce immune-suppressive, teratogenic, carcinogenic, and mutagenic effects [12]; and affect and damage the nervous system, causing neurodegenerative and hepatorenal disorders [13,14]. Hence, there is a need to develop a valid and reliable method for the simultaneous detection of multiple pesticide residues on multiple herbs, and to detect and evaluate the associated risks. As medicinal plants contain complex active ingredients in addition to pigments, it is a challenging task to develop a method for the detection of multi-pesticide residues in Chinese herbs. Complex components in TCMs can interfere with the extraction and separation of target analytes; therefore, few methods have been established thus far for the simultaneous determination of multiple pesticide residues in a variety of TCMs.

Currently, the detection methods applied to pesticide residues on TCMs include liquid chromatography-tandem mass spectrometry (LC-MS/MS) [15], gas chromatography-tandem mass spectrometry (GC-MS/MS) [16], high performance liquid chromatography-tandem mass spectrometry (HPLC-MS/MS) [17], and ultra-performance liquid chromatography-tandem mass spectrometry (UPLC-MS/MS) [18]. The key to being able to accurately quantify the target analyte during the assay is the sample pretreatment process. QuEChERS is a quick, easy, cheap, effective, rugged, and safe sample pretreatment method developed by Anastassiades et al. in 2003 [19]. This method was initially proposed for pesticide residue detection in fruits and vegetables, and has been cited and optimized by more researchers. Therefore, this method can be applied to a wide range of substrates and is currently used as the first choice for rapid pretreatment technology. Wu et al. [8] used the QuEChERS method to establish the LC-MS/MS method for the detection of 114 pesticide residues in different *Fritillaria* species by optimizing the extraction solvent and sorbent amount. Yang et al. [18] optimized the extraction time and adsorbent ratios to establish a multi-residue analysis method for 249 pesticides in Panax notoginseng. Sutcharitchan et al. [20] used HPLC-MS/MS to optimize the various conditions of QuEChERS and established a detection method for 39 plant growth regulators in Dangshen and Sanqi. The methods noted above mostly focus on one, or one class of, herbal medicine, and thus cannot meet the requirements of rapid and high throughput detection covering various types of herbal medicines of plant origin. In addition, the adsorbents used are commonly ethylenediamine-N-propylsilane (PSA), grinded carbon black (GCB), and octadecylsilane (C18), which are not enough to remove pigments and serious matrix interference. However, multi-walled carbon nanotubes (MWCNTs) have been frequently applied to new adsorbents in recent years due to their large surface area, unique structure, and easy derivatization [21]. In this study, we took *F. thunbergii*, *C. morifolium*, and *D. officinale* as the research objects, representing the plant-derived Chinese herbal medicines of bulbs, stems and flowers, respectively, and focused on analyzing the adsorption performance of MWCNTs and their derivatives, simplifying the adsorbent types, and establishing an efficient multi-residue analysis method.

In this study, HPLC-MS/MS combined with QuEChERS was used to develop a simple, rapid, accurate, and efficient method for the detection of 108 pesticides in three kinds of TCMs. Sixty batches of *F. thunbergii*, *C. morifolium*, and *D. officinale* were tested for pesticide residues, and the pesticide residues were monitored. In order to improve the detection efficiency and quickly establish the detection method for multiple pesticide residues on three kinds of TCMs, this study aimed to mix the three samples evenly and then take samples, optimize the pretreatment method based on the mixed samples, and validate the optimized method on a single sample. Finally, 108 pesticide detection results were obtained for *F. thunbergii*, *C. morifolium*, and *D. officinale*. The main purpose of this study is to simplify the pretreatment optimization process, which can not only save time and cost, but also avoid the waste of resources. This study can also provide ideas for the development of other pesticide residue detection methods for various samples.

## 2. Results and Discussion

### 2.1. MS/MS Conditions Optimization

Mass spectrometry parameter optimization is an important step in obtaining maximum sensitivity [22]. In this study, the standard was prepared as a 1 mg/L methanol solution, and two scanning modes of positive and negative ions were established at the same time. A first-level full scan was performed for each single standard solution of the analytes to be measured to obtain the parent ions with high and stable abundance. After determining the parent ion, the fragmentor of the compound was optimized in single ion monitoring (SIM) mode, in which the parent ion enters the secondary mass spectrum and undergoes reactions such as breakage or rearrangement to produce different ion fragments. In product ion mode, a certain amount of collision energy (CE) was applied to the parent ion of the compound to obtain the corresponding ion fragmentation. Finally, in the multiple reaction monitoring (MRM) mode, the optimal collision energy of the target ion fragment is optimized to obtain the best mass spectrometry parameters. The optimal MRM detection parameters for each pesticide are listed in Appendix A.

### 2.2. Optimization Chromatographic Conditions

After determining the optimal MRM parameters, the HPLC method was optimized to enable fast and reliable separation of the peaks of the target analytes. Experiments were conducted to compare C18 and Hilic columns for the separation of the analytes to be measured. Since the C18 column better retained the analyte and exhibited a better peak shape and response, the C18 column was chosen for analysis of the analyte. Then, the mobile phases of 0.2% formic acid (FA) + 2 mM ammonium formate in water, and 0.2% FA + 2 mM ammonium formate in methanol, were compared with the mobile phases of 2 mM ammonium formate in water and methanol. It was found that the peak shape of each pesticide was better when the mobile phase was the latter. Subsequently, the injection volumes were compared, and higher sensitivity was obtained with larger injection volumes, and better peak shapes were obtained with smaller injection volumes. In this experiment, the peak shapes and sensitivities of each target analyte were compared at 1, 2, and 5 μL. It was found that some of the target analytes showed flat-topped and pre-delayed peaks at the injection volume of 5 μL. The peak shapes of all analytes were good and similar at 1 and 2 μL injection volumes. In order to balance the dual requirements of high sensitivity and optimal peak shape, 2 μL was chosen as the injection volume. Figure 1 shows the effect of injection volume on peak shape for six representative pesticides at 2 and 5 μL.

### 2.3. Extraction Method Optimization

When developing a sample pretreatment method, following conventional analytical methods not only increases the workload, but also consumes additional reagents. We tried to mix the dry samples of three samples together and use the mixed samples as a matrix. This matrix was used as the research object for the development and optimization of each step to quickly establish a multi-residue detection method. Finally, the method was validated separately for each of the three herbs to determine the applicability and reliability of the developed method. The optimization of the pretreatment method according to this method will be more effective with less effort. Therefore, we weighed the same amount of the three samples into a beaker, mixed the samples thoroughly, and then weighed 2 g into a 50 mL centrifuge tube for subsequent method development. The QuEChERS method was originally developed for fruits and vegetables, which have a certain moisture content. However, because herbs are dried and have a moisture content of <10%, this method cannot be directly used; thus, we chose to add a certain volume of water in order to keep the sample in a moist state [23]. A sample volume of 2 g was observed to account for approximately 5 mL of the 50 mL centrifuge tube volume; therefore, 5 mL of water was chosen to be added to moisten the sample [24]. We directly disregarded methanol as an extraction solvent because of its high polarity, which may inhibit the complete extraction of non-polar or medium-polarity pesticides. In addition, methanol as an extraction solvent would also extract a large amount of sugar, making the extraction solution dark and viscous [25]. We chose to use acetonitrile as the extraction solvent to examine the effect of the type of acid in acetonitrile (acetonitrile, acetonitrile with 0.1% FA, acetonitrile with 0.1% acetic acid (HAC)) on the extraction efficiency, as the addition of acid increases the recovery of acidic PGRs [20]. After determining the extraction solvent, the best extraction method was obtained by varying the volume of the extraction solution (10 and 20 mL). The best conditions were determined by the mean recovery and relative standard deviation (RSD).

The results are shown in Figure 2. Under the condition of 0.1% FA acetonitrile, the recoveries of four pesticides were less than 70%, that is, indole propamocarb (67.60%), 6-furfuryl aminopurine (67.87%), brassinolide (58.41%), and carbosulfan (69.8%). Under the condition of 0.1% HAC acetonitrile, the recoveries of three pesticides were less than 70%: propamocarb (67.40%), 6-furfuryl aminopurine (65.64%), and brassinolide (57.34%). Under the condition of acetonitrile as the extraction solvent, all recoveries were 70% or higher, even though some pesticide recovery under acid conditions is better. However, in order to consider all pesticide recoveries, we finally chose acetonitrile as the extraction solvent. In the comparison of 10 and 20 mL of extraction solvent, it was found that the average recovery rate of 108 pesticides was 97.33% with RSD of 3.99 under the condition of 10 mL, and the average recovery rate of all pesticides was 94.72% with RSD of 3.40 under the condition of 20 mL of acetonitrile. We also performed a significance analysis, and the difference between 10 and 20 mL acetonitrile volumes showed a significant difference at the 0.05 significance level; thus, 10 mL acetonitrile was finally chosen as the best extraction condition.

### 2.4. Clean Up Method Optimization

In the step of screening adsorbents, we referred to the method of Wang et al. [26]. We first conducted a series of simulated adsorption tests with solvent standard solutions to evaluate the adsorption behavior of various adsorbents on pesticides. The adsorbents in Appendix A were dispersed in 1.6 mL of a mixed standard solution containing all the target pesticides prepared in methanol at a concentration of 0.5 mg/L, and vortexed for 1 min to allow complete interaction between the target pesticides and the adsorbents. The supernatant was then centrifuged and separated, and the supernatant was passed through a 0.22 μm filter membrane for analysis to determine the amount of non-adsorbed pesticide. The adsorption capacity of the various adsorbents for the target pesticide was evaluated by comparing the mass ratio of non-adsorbed pesticide to the original added pesticide, with the results requiring ≥70%. This indicates that this adsorbent has a weak adsorption capacity for the target pesticide and can ensure satisfactory recovery during dispersive solid-phase extraction (d-SPE) decontamination of the matrix. We calculated the recovery of each target pesticide for all combinations of sorbents, and the results are shown in Figure 3. ZrO_2_ had the highest number of pesticides with ≥70% recovery at 105, followed by PSA and C18 at 104 and 103 respectively; the different multi-walled carbon nanotubes and their derivatizations showed stronger adsorption than the PSA and C18 combinations. We speculate that this may be a result of the preferential adsorption of pesticides by multi-walled carbon nanotubes in the absence of matrix interference, resulting in low recovery. Several representative groups of sorbents were selected for application to *D. officinale* base extracts to compare the purification effect, as *D. officinale* extracts were the darkest of the three substrates. From Figure 4, we observed that the cleanest purification was in group 7 (MWCNTs-NH_2_), and group 11 (PSA + C18) had the least satisfactory purification. Although ZrO_2_ has the best recovery, it was inferior to MWCNTs in terms of purification effect.

To investigate the adsorption effect of multi-walled carbon nanotubes on matrix interference, six groups from the above combinations were selected for matrix adsorption experiments. The experimental results are shown in Figure 5. The effect of MWCNTs on pesticide recovery in the presence of matrix interference showed opposite results to those described above. These results were all higher than those from the combination of PSA + C18, with a minimum average recovery of 79.09%. This result verifies that in the presence of matrix interference, MWCNTs bind preferentially to impurities, exhibiting better purification and higher yields. We also found that shorter lengths of g-MWCNTs showed better recovery than longer lengths of g-MWCNTs with the same outer diameter. For five different derivative MWCNTs, group 7 (MWCNTs-NH_2_) had the highest mean recovery rate of 96.56%, with 104 pesticides having recoveries ≥70%. Recovery failed for only four pesticides, namely, betaine (54.22%), acephate (63.05%), 6-Kinetin (52.37%), and 6-Benzylaminopurine (64.20%). The average recovery of group 8 (MWCNTs-OH) was second only to that of MWCNTs-NH_2,_ at 95.83%, except for the four pesticides mentioned above, which did not have satisfactory recoveries, and thidiazuron, with an average recovery of 69.40%. In summary, MWCNTs-NH_2_ was considered to be the most suitable absorbent.

Finally, the dose of MWCNTs-NH_2_ was further optimized to compare the effect of the three doses on the recovery of the target pesticides (Figure 5B). As the dose increased, the purification effect was better, but 20 mg of MWCNTs-NH_2_ was also more potent for pesticide adsorption, with only 98 pesticides recovering more than 70%, at an average recovery rate of 90.86%. However, the average recovery rate for all pesticides at 5 mg was 89.56%. This was possibly due to the fact that low doses were not sufficient to purify the interfering components in 2 g samples, resulting in greater matrix interference and yielding an average recovery of MWCNTs-NH_2_ (95.52%) below 10 mg. Therefore, 10 mg of MWCNTs-NH_2_ was selected as the optimal dose of sorbent.

### 2.5. Matrix Effect

Matrix effects are caused by the co-elution of matrix constituents. This affects the ionization of the target compound, leading to ion enhancement or inhibition [27]. To determine the matrix effect, we compared the slope of the standard curve obtained from a matrix standard solution with that obtained from a solvent-based standard solution [28]. The matrix effects of the target compounds in *F. thunbergii*, *C. morifolium*, and *D. officinale* were obtained, and are listed in Appendix A. The matrix effects were negligible in the range of −20% to +20%; values higher than +20% resulted in matrix enhancement; and those lower than −20% resulted in matrix inhibition. Among the three herbs, most of the pesticides showed matrix enhancement in *F. thunbergii*, and matrix inhibition in *C. morifolium* and *D. officinale*. The standard matrix-matching calibration approach was used in pesticide residue analysis to improve the accuracy of the quantitative results.

### 2.6. Method Validation

The above optimal pretreatment method was developed for the study of mixed samples in order to improve efficiency. This method was designed to be used on a single matrix, so the three herbs were validated separately during method validation. Both matrix-matched standard solutions and solvent standard solutions were prepared at concentrations of 2, 5, 10, 20, 50, 100, and 200 μg/L. The accuracy of the method was verified by three spiking levels, high, medium, and low (100, 50, and 20 μg/kg, respectively), with five replicates at each level. The fitted linear regression equations, limit of detection (LOD), and limit of quantification (LOQ) are presented in Appendix A for the samples of *F. thunbergii*, *C. morifolium*, and *D. officinale*, respectively. The recovery and RSD of each target pesticide are shown in Appendix A. In all three TCMs, the R^2^ of the calibration curves for all target pesticides was higher than 0.9984, with excellent linearity. The precision (based on RSD values) was in the ranges of 0.27–16.82% (intra-day) and 1.21–11.61% (inter-day) for *F. thunbergii*, 0.56–14.21% (intra-day) and 0.65–15.03% (inter-day) for *C. morifolium*, and 0.13–12.58% (intra-day) and 0.67–11.91% (inter-day) for *D. officinale*. This shows that the method has high sensitivity and meets the regulatory requirements. In *F. thunbergii*, the recoveries of pesticides (95%) excluding betaine, 6-kinetin, 6-benzylaminopurine, thidiazuron, and forchlorfenuron were in the 71.37–109.49% range. In *C. morifolium*, excluding betaine, propamocarb, 6-kinetin, 6-benzylaminopurine, and carbosulfan, the recoveries of the pesticides (95%) were in the 70.54–114.32% range. In *D. officinale*, except for 6-kinetin, 6-benzylaminopurine, thidiazuron, and carbosulfan, the recoveries of the pesticides (96%) were in the 70.17–114.41% range. The RSD values of all pesticides were less than 20%.

### 2.7. Application to Real Samples

Qualitative and quantitative tests of pesticide residues were performed on commercially purchased samples using the developed QuEChERS pretreatment method. The screening of 108 pesticide residues was performed on 20 samples each of *F. thunbergii*, *C. morifolium*, and *D. officinale*. Appendix A show that significant quantities of pesticides were also used in the cultivation of herbal medicines. Pesticide residues were detected in all 60 samples in this study, with a detection rate of 100%. Overall, 72 residues were detected across all samples: 30 insecticides, 27 fungicides, 13 plant growth regulators, and 2 herbicides (Figure 6A). There were 2, 2, and 17 pesticides with a 100% detection rate in *F. thunbergii*, *D. officinale*, and *C. morifolium*, respectively. In *D. officinale*, in the largest number of samples, 5–10 pesticides were detected at the same time, accounting for 45% of the total number of samples; in *F. thunbergii*, in the largest number of samples, 11–15 pesticides were detected at the same time, accounting for 45% of the total number of samples; in *C. morifolium*, the lowest number of pesticides that could be detected at the same time in a single sample was 32, and the number of pesticides detected was higher than in the other two types of Chinese herbs (Figure 6A). It is possible that because flowers are edible parts of the plant, and are more susceptible to pests and diseases, growers use more pesticides to increase yields. Currently, in the National Food Safety Standards-Maximum Residue Limits for Pesticides in Food (GB 2763-2021), only imidacloprid, thiameth-oxam, dimethomorph, carbosulfan, carbofuran, dimethoate, acephate and difenoconazole have residue requirements for TCM (Appendix A). In the three herbs, the pesticides detected did not exceed the limit values specified in GB 2763-2021.

## 3. Materials and Methods

### 3.1. Chemicals and Reagents

All pesticide standards were purchased from Shanghai Anpu Experimental Technology Co. (Shanghai, China). HPLC grade acetonitrile and methanol were obtained by Merck Co. (Darmstadt, Germany). HPLC grade formic acid was purchased from Anaqua Chemicals Supply (Wilmington, NC, USA). Anhydrous magnesium sulfate (MgSO_4_) was obtained from RHAWN (Shanghai, China). MWCNT (O.D. 20–30 nm, L. 0.5–2 μm, O.D. 10–20 nm, L. 0.5–2 μm, O.D. 10–20 nm, L. 10–30 μm) and g-MWCNTs (O.D. 10–20 nm, L. 5–30 μm) were obtained by XFNANO (Nanjing, China). g-MWCNTs (O.D. 30–50 nm, L. 10–20 μm), MWCNTs-COOH (O.D. 30–50 nm, L. 10–20 μm), MWCNTs-OH (O.D. 30–50 nm, L. 10–20 μm), MWCNTs-NH_2_ (O.D. 8–15 nm, L. −50 μm), and sorbents ZrO_2_ were purchased from Aladdin Industrial Corporation (Shanghai, China). PSA and C18 were obtained by Agilent Technologies (Tianjin, China). 2 mL Agela Cleanert MAS-Q purification tubes (PSA, 50 mg; C18, 50 mg; MgSO_4_, 150 mg) were purchased from Bonna-Agela Technologies Co. (Tianjin, China). Sodium chloride (NaCl) and AR-acetonitrile were purchased from Shanghai Lingfeng Chemical Reagent Co. (Shanghai, China).

A separate standard solution for each pesticide was prepared in acetonitrile at a concentration of 1000 mg/L. Then, the mixed standard solutions were prepared in methanol. All standard solutions were stored in a refrigerator at 4 °C for use in subsequent experiments.

### 3.2. HPLC-MS/MS Conditions

Ultra-high performance liquid chromatography was coupled with tandem mass spectrometry (HPLC-MS/MS, LCMS 8050, Shimadzu, Japan). The LC was equipped with a Waters ACQUITY UPLC^®^ BEH C18 column (2.1 mm × 100 mm, 1.7 μm, Phenomenex, Torrance, CA, USA). The column temperature was 40 °C. The mobile phase was 2 mM ammonium formate aqueous solution (A) and pure methanol (B) with a flow rate of 0.2 mL/min and an injection volume of 2 μL. Using the linear gradient elution method, eluent B was increased from 10% to 50% within 0.5–5 min, and to 90% within 5–15 min; the gradient was maintained at 5 min, then decreased to 10% within 20–20.1 min, and rebalanced at 10% within 20.1–24 min.

Mass spectrometry analysis was performed using an MRM model, ionized by electrospray ionization (ESI), with both positive and negative ion patterns. The flow rates of nitrogen (N_2_, 99.95%) as the nebulizer gas and as the drying gas were 3.0 and 10.0 L/min, respectively; the collision gas was 99.99% argon with a pressure of 270 kPa; the heated gas flow rate was 10.0 L/min. Other parameters were as follows: desolvation line temperature of 250 °C, interface voltage of 4.0 kV, interface temperature of 300 °C, detector voltage of 1.82 kV. Optimized retention time, parent and daughter ions, and collision energy (CE) for each target compound are listed in Appendix A.

### 3.3. Sample Pretreatment

The blank samples of *F. thunbergii*, *C. morifolium*, and *D. officinale* came from Zhejiang province. Dry samples were ground directly into powder, and fresh samples were dried at 60 °C before being ground into powder. Samples were sealed and stored at room temperature.

Twenty servings of each of the three herbs were purchased from local pharmacies and online stores, ground into powder and stored at room temperature for actual sample testing.

### 3.4. Sample Preparation

After weighing 2 g of sample into a 50 mL centrifuge tube, 5 mL H_2_O was added to wet the sample; 10 mL of analytically pure acetonitrile was added and shaken for 10 min, followed by 3 g NaCl. The mixture was shaken for 1 min and centrifuged at 4000 rpm for 5 min; then, 1.6 mL of supernatant was transferred to an EP tube containing 10 mg MWCNTs-NH_2_ and 150 mg MgSO_4_, vortexed for 1 min, and centrifuged at 10,000 rpm for 5 min. Finally, the supernatant was passed through a 0.22 μm nylon syringe filter into a vial, and then analyzed using HPLC-MS/MS.

### 3.5. Method Validation

According to the analytical quality control standards of SANTE/11813/2017 guidelines [29], the validation of the developed method was carried out using the following parameters: linearity, matrix effect, accuracy and precision, limit of quantification (LOQ), and limit of detection (LOD). Matrix-matched and solvent standard curves were constructed using concentration as the horizontal coordinate and peak area as the vertical coordinate, and the correlation coefficient (R^2^) of the curves was used to estimate the linearity. The accuracy of the method was assessed by performing spiked recovery experiments on blank samples. RSD was used for the assessment of intra-day precision and inter-day precision. The LOD of the proposed method was calculated at a signal-to-noise ratios (S/N) of 3. The LOQ of the proposed method was calculated at a signal-to-noise ratios (S/N) of 10. The matrix effect (*Me*) was obtained by comparing the slope of the matrix calibration curve to that of the solvent calibration curve, calculated using the following formula:(1)$$Me(\%)=\left(\frac{S_{matrix}}{S_{solvent}}-1\right)\times 100$$
where $S_{matrix}$ is the slope of the matrix-matching calibration curve, and $S_{solvent}$ is the slope of the solvent calibration curve.

### 3.6. Statistical Analysis

Results are expressed as the means ± standard deviation (SD). The recovery data of pesticides under different extraction solvent volumes were statistically analyzed by GraphPad prism 9.0. One-way analysis of variance (ANOVA) was performed at *p* < 0.05. Histograms and pie chart were plotted with GraphPad prism 9.0 and Origin 2020. The radar map was generated by Origin 2020.

## 4. Conclusions

In this study, a fast, simple, efficient, robust, and high-throughput HPLC-MS/MS method was developed for the multi-residue analysis of 108 pesticides in the frequently used TCMs of *F. thunbergii*, *C. morifolium*, and *D. officinale* by optimizing pretreatment of the mixing sample. The conditions of mass spectrometry and liquid phase were optimized, and the method combined with an ESI source in MRM mode allowed the simultaneous quantification of 108 pesticides in Chinese herbal medicines of complex composition within 24 min. The sample preparation process was optimized based on the previous QuEChERS method for the selected three TCMs. After method optimization, we concluded that satisfactory recoveries were obtained via extraction in 10 mL acetonitrile for 10 min and d-SPE cleanup, with 150 mg MgSO_4_ in combination with 10 mg MWCNTs-NH_2_. The method was also comprehensively validated and proved to be highly sensitive, precise, and accurate. It was also applied to the analysis of 60 batches of real samples. Pesticide residue analysis methods require simple and efficient pretreatment methods. Thus, the improved method developed in this study can be applied to the detection of pesticide residues in similar Chinese medicines or in samples of other properties that are related to these products.

## Figures and Tables

**Figure 1 molecules-27-07636-f001:**
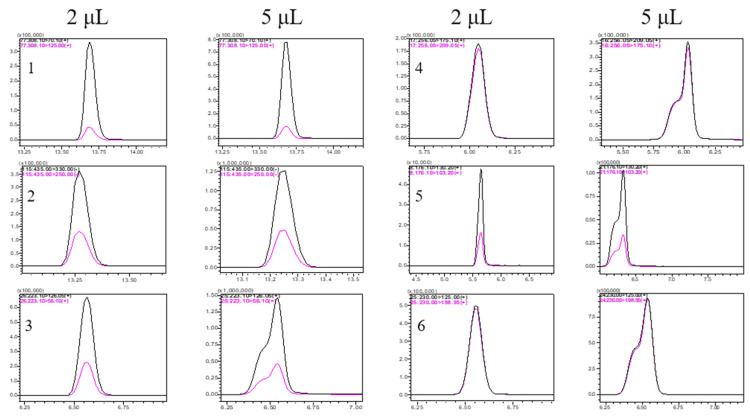
Chromatograms of six representative pesticides (Tebuconazole (1), Fipronil (2), Acetamiprid (3), Imidacloprid (4), 3-Indoleacetic Acid (5), Dimethoate (6); injection volume: 2 and 5 μL.

**Figure 2 molecules-27-07636-f002:**
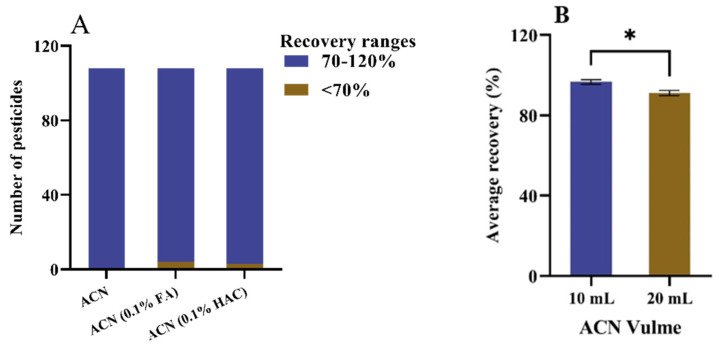
(**A**) Influence of extraction solvent on the recovery of the pesticides from the spiked 100 μg/kg mixed sample. (**B**) Influence of extraction solvent volume on the average recovery of the pesticides from the spiked 100 μg/kg mixed sample. “*” means that there was a significant difference between the groups of data at *p* level < 0.05.

**Figure 3 molecules-27-07636-f003:**
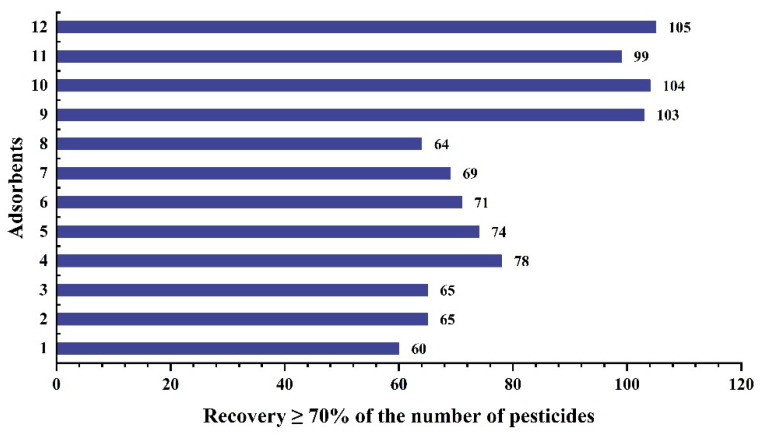
The number of pesticides with recovery ≥70% in methanol among adsorbent groups.

**Figure 4 molecules-27-07636-f004:**
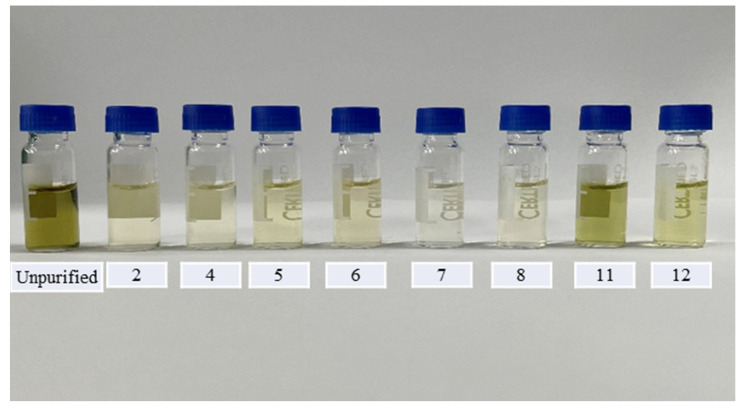
The adsorption effects of eight different adsorption agents for purifying the interfering components in *D. officinale*. (2) 10 mg MWCNT (short) + 50 mg MgSO_4_, (4) 10 mg g-MWCNTs + 150 mg MgSO_4_, (5) 10 mg g-MWCNTs + 150 mg MgSO_4_, (6) 10 mg MWCNTs-COOH + 150 mg MgSO_4_, (7) 10 mg MWCNTs-NH_2_ + 150 mg MgSO_4_, (8) 10 mg MWCNTs-OH + 150 mg MgSO_4_, (11) 50 mg PSA + 50 mg C18 + 150 mg MgSO_4_, (12) 10 mg ZrO_2_ + 150 mg MgSO_4_.

**Figure 5 molecules-27-07636-f005:**
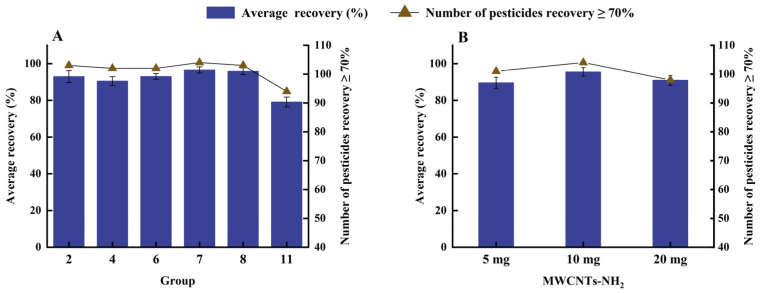
(**A**) Influence of the absorbent on the recoveries of pesticides from the spiked 100 μg/kg mixed sample. (**B**) Influence of MWCNTs-NH_2_ dosages on the recoveries of pesticides from the spiked 100 μg/kg mixed sample.

**Figure 6 molecules-27-07636-f006:**
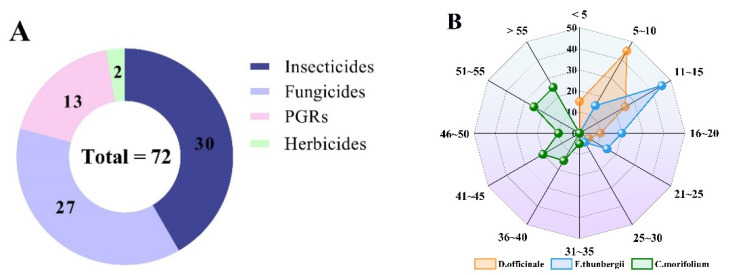
(**A**) Distribution of pesticide species tested in three TCM real samples. (**B**) Number of detectable residues in individual samples and the sample proportion.

## Data Availability

Not applicable.

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
