# Peer review of "Simultaneous Determination of 108 Pesticide Residues in Three Traditional Chinese Medicines Using a Modified QuEChERS Mixed Sample Preparation Method and HPLC-MS/MS"

_molecules, 2022, doi:10.3390/molecules27217636_

Round 1
Reviewer 1 Report
General comments
This research built and developed a method for simultaneous detection of 108 pesticides in three Traditional Chinese Medicines, and it was applied well. This study is very promising in the detection of multiple pesticide residue species in Traditional Chinese Medicines. I am rather in favor of publishing the present work, however, there are minor issues in the text that need to be revised.
Specific comments
1. In the abstract, RSD appears for the first time, please spell it fully
2. In keywords, “traditional chinese medicines” should be changed into “traditional Chinese medicines”.
3. Page 2,line 86,he first appearance of C18 shall indicate its full name " octadecylsilane
4. Page 4, line 141, 2 μl and 5 μl should be changed to 2 μL and 5 μL, please check the whole manuscript for corrections.
5. Page 5,figure 2, the data presented in the manuscript be presented as mean ± SD as much as possible.
6. In the section 3. Materials and Methods, the method description should be in the past tense.
7. Line 20 is "LOQs" and line 368 is "LOQ". Please correct them in the whole manuscript.
8. Please add the description of statistical analysis of data in the method and material section.
9. Line 121, whether fragmen are misspelled, and the corrected is " fragment"?
10. Line 194-196, “In the step of screening the adsorbents, we refer to the method of Wang et al., and first conduct a series of simulated adsorption tests with solvent standard solutions to evaluate the adsorption behavior of various adsorbents on pesticides [26].” should be changed to “In the step of screening adsorbents, we referred to the method of Wang et al. and first conducted a series of simulated adsorption tests with solvent standard solutions to evaluate the adsorption behavior of various adsorbents on pesticides [26].”
10. In the section 2. Results and Discussion, the description of the results should be in the past tense or past perfect tense. Please correct it.
11. line 332, “℃”should be changed into “℃”, please checked the whole text.
Author Response
- In the abstract, RSD appears for the first time, please spell it fully.
Response: Thank you very much for your helpful suggestion. In the abstract, “RSD” was revised to “relative standard deviation (RSD)”.
- In keywords, "traditional chinese medicines" should be changed into "traditional Chinese medicines".
Response: Thank you very much for your helpful suggestion. "traditional chinese medicines" was changed to "traditional Chinese medicines".
- Page 2, line 86, he first appearance of C18 shall indicate its full name " octadecylsilane?
Response: We apologize for our negligence. In the revised manuscript, we fully explained C18.
Line 89-92: "And the adsorbents used are common ethylenediamine-N-propylsilane (PSA), grinded carbon black (GCB) and octadecylsilane (C18), which are not enough to remove pigments and serious matrix interference."
- Page 4, line 141, 2 μl and 5 μl should be changed to 2 μL and 5 μL, please check the whole manuscript for corrections.
Response: Thank you for your reminding. “2 μl and 5 μl” has been changed to “2 μL and 5 μL”.
- Page 5, figure 2, the data presented in the manuscript be presented as mean ± SD as much as possible.
Response: Thank you very much for your helpful suggestion. The meaning of our original expression was not very clear, in figure 2A, the X-axis is the number of pesticides that cannot be presented with mean ± SD. So we have made simple changes to make it easier for readers to observe. Meanwhile, the mean ± SD was added to the Figure 5.
- In the section 3. Materials and Methods, the method description should be in the past tense.
Response: Thank you very much for your helpful suggestion. The sentence tenses of “Materials and Methods (section 3)” were checked and corrected in the revised manuscript.
- Line 20 is "LOQs" and line 368 is "LOQ". Please correct them in the whole manuscript.
Response: Thank you very much for your helpful suggestion. The "LOQs" was revised to "LOQ".
- Please add the description of statistical analysis of data in the method and material section.
Response: Your suggestion made the manuscript more complete, so we add "3.6 Statistical analysis" in the revised manuscript to describe statistical analysis and mapping software.
- Line 121, whether fragmen are misspelled, and the corrected is " fragment"?
Response: Thank you for your reminding. The word "fragmen" has been changed to "fragment".
- Line 194-196, "In the step of screening the adsorbents, we refer to the method of Wang et al., and first conduct a series of simulated adsorption tests with solvent standard solutions to evaluate the adsorption behavior of various adsorbents on pesticides [26]." should be changed to "In the step of screening adsorbents, we referred to the method of Wang et al. and first conducted a series of simulated adsorption tests with solvent standard solutions to evaluate the adsorption behavior of various adsorbents on pesticides [26]."
Response: Thank you very much for your helpful suggestion. The sentence tense has been changed "In the step of screening adsorbents, we referred to the method of Wang et al. [26] and first conducted a series of simulated adsorption tests with solvent standard solutions to evaluate the adsorption behavior of various adsorbents on pesticides."
- In the section 2. Results and Discussion, the description of the results should be in the past tense or past perfect tense. Please correct it.
Response: Thank you very much for your helpful suggestion. The sentence tenses of “Results and Discussion (section 2)” were checked and corrected in the revised manuscript.
- line 332, “℃” should be changed into “℃”, please checked the whole text.
Response: Thank you for your reminding. All "℃" formats have been corrected in the revised manuscript.
Reviewer 2 Report
The paper is well-written and thoroughly detailed.
I believe the authors should reconsider keeping the following lines (72-75, 82-89, 147-150) as once sentence only, and should change it to two separate sentences, because they seem too long:
"QuEChERS is a quick, easy, cheap, effective, rugged, and safe sample pre-treatment 72 method developed by Anastassiades et al. in 2003 [19], the method was initially used for 73 the detection of pesticide residues in fruit and vegetables, but has since been cited and 74 optimised by more researchers and applied to a wider range of assays and matrices, be- 75 coming the rapid pretreatment technique of choice for matrices."
"The above developed 82 methods mostly focus on one or one class of herbal medicines, which cannot meet the 83 requirements of rapid and high throughput detection covering various types of herbal 84 medicines of plant origin, and the adsorbents used are common ethylenediamine-N- 85 propylsilane (PSA), grinded carbon black (GCB) and C18, which are not enough to remove 86 pigments and serious matrix interference, and Multi Walled Carbon Nanotubes 87 (MWCNTs) have become the high-frequency use objects of new adsorbents in recent years 88 due to their large surface area, unique structure and easy of derivatization [21]."
"Before developing the sample pre-treatment method, considering the number of pes- 147 ticides and the types of samples, if we follow the traditional analytical method and apply 148 each of the conditions we want to optimize to all three samples, it will greatly increase the 149 workload, not only will it take more time, but also more reagents will be consumed."
I also appreciate the brevity of the journal article.
Author Response
I believe the authors should reconsider keeping the following lines (72-75, 82-89, 147-150) as once sentence only, and should change it to two separate sentences, because they seem too long:
- "QuEChERS is a quick, easy, cheap, effective, rugged, and safe sample pre-treatment 72 method developed by Anastassiades et al. in 2003 [19], the method was initially used for 73 the detection of pesticide residues in fruit and vegetables, but has since been cited and 74 optimised by more researchers and applied to a wider range of assays and matrices, be- 75 coming the rapid pretreatment technique of choice for matrices."
Response: We agree with your suggestion very much. We have divided the sentence into three sentences in the revised manuscript.
Line 73-80: "QuEChERS is a quick, easy, cheap, effective, rugged, and safe sample pretreatment meth-od developed by Anastassiades et al. in 2003 [19]. This method was initially proposed for pesticide residue detection in fruits and vegetables, and has been cited and optimized by more researchers. Therefore, this method can be applied to a wide range of substrates and is currently used as the first choice for rapid pretreatment technology."
- "The above developed 82 methods mostly focus on one or one class of herbal medicines, which cannot meet the 83 requirements of rapid and high throughput detection covering various types of herbal 84 medicines of plant origin, and the adsorbents used are common ethylenediamine-N- 85 propylsilane (PSA), grinded carbon black (GCB) and C18, which are not enough to remove 86 pigments and serious matrix interference, and Multi Walled Carbon Nanotubes 87 (MWCNTs) have become the high-frequency use objects of new adsorbents in recent years 88 due to their large surface area, unique structure and easy of derivatization [21]."
Response: Based on your suggestion, we have divided the sentence into three sentences in the revised manuscript.
Line 87-94: "The above developed methods mostly focus on one or one class of herbal medicines, which cannot meet the requirements of rapid and high throughput detection covering various types of herbal medicines of plant origin. And the adsorbents used are common ethylenediamine-N-propylsilane (PSA), grinded carbon black (GCB) and octadecylsilane (C18), which are not enough to remove pigments and serious matrix interference. However, Multi Walled Carbon Nanotubes (MWCNTs) have become the high-frequency applications of new adsorbents in recent years due to their large surface area, unique structure and easy of derivatization [21]."
- "Before developing the sample pre-treatment method, considering the number of 147 pesticides and the types of samples, if we follow the traditional analytical method and apply 148 each of the conditions we want to optimize to all three samples, it will greatly increase the 149 workload, not only will it take more time, but also more reagents will be consumed."
Response: We have rephrased this sentence to "When developing sample pretreatment method, following conventional analytical methods not only increases the workload, but also consumes additional reagents."